# Proteo-Molecular Investigation of Cultivated Rice, Wild Rice, and Barley Provides Clues of Defense Responses against *Rhizoctonia solani* Infection

**DOI:** 10.3390/bioengineering9100589

**Published:** 2022-10-20

**Authors:** Md. Shamim, Divakar Sharma, Deepa Bisht, Rashmi Maurya, Mayank Kaashyap, Deepti Srivastava, Anurag Mishra, Deepak Kumar, Mahesh Kumar, Vijaya Naresh Juturu, N. A. Khan, Sameer Chaudhary, Raja Hussain, K. N. Singh

**Affiliations:** 1Department of Plant Molecular Biology and Genetic Engineering, A.N.D. University of Agriculture and Technology, Kumarganj, Ayodhya 224229, Uttar Pradesh, India; 2Department of Molecular Biology and Genetic Engineering, Dr. Kalam Agricultural College, Bihar Agricultural University, Sabour, Bhagalpur, Arrabari, Kishanganj 855107, Bihar, India; 3Department of Microbiology, Maulana Azad Medical College, Bahadur Shah Zafar Marg, New Delhi 110002, Delhi, India; 4Department of Biochemistry, ICMR-National JALMA Institute for Leprosy and Other Mycobacterial Diseases, Tajganj, Agra 282001, Uttar Pradesh, India; 5School of Computational and Integrative Sciences, Jawaharlal Nehru University, New Delhi 110067, Delhi, India; 6School of Life Science, RMIT University, Bundoora, Melbourne, VIC 3083, Australia; 7Department of Agriculture, Integral Institute of Agricultural Science and Technology, Integral University, Lucknow 226026, Uttar Pradesh, India; 8Department of Manufacturing and Development, Nextnode Bio Science, Pvt. Ltd., Kadi 384440, Gujarat, India; 9Agri Biotech Foundation, Formerly A P Netherlands Biotechnology, Programme, Rajendra Nagar, Hyderabad 500030, Telangana, India; 10RASA Life Science Informatics, Law College Road, Pune 411052, Maharashtra, India

**Keywords:** *Rhizoctonia solani*, pathogenesis, cereal crops, proteomics

## Abstract

*Rhizoctonia solani* is a soil-borne fungus causing sheath blight disease in cereal crops including rice. Genetic resistance to sheath blight disease in cereal crops is not well understood in most of the host(s). Aside from this, a comparative study on the different hosts at the biochemical and proteomic level upon *R. solani* infection was not reported earlier. Here, we performed proteomic based analysis and studied defense pathways among cultivated rice (cv. Pusa Basmati-1), wild rice accession (*Oryza grandiglumis*), and barley (cv. NDB-1445) after inoculation with *R. solani*. Increased levels of phenol, peroxidase, and β-1, 3-glucanase were observed in infected tissue as compared to the control in all of the hosts. Wild rice accession *O. grandiglumis* showed a higher level of biochemical signals than barley cv. NDB 1445 and cultivated rice cv. Pusa Basmati-1. Using two-dimensional polyacrylamide gel electrophoresis (2D-PAGE) and mass spectrometry (MS), differently expressed proteins were also studied in control and after inoculation with *R. solani*. Wild rice accession *O. grandiglumis* induced a cysteine protease inhibitor and zinc finger proteins, which have defense functions and resistance against fungal pathogens. On the other hand, barley cv. NDB-1445 and cultivated rice cv. Pusa Basmati-1 mainly induce energy metabolism-related proteins/signals after inoculation with *R. solani* in comparison to wild rice accession *O. grandiglumis*. The present comprehensive study of *R. solani* interaction using three hosts, namely, Pusa Basmati-1 (cultivated rice), *O. grandiglumis* (wild rice), and NDB-1445 (barley) would interpret wider possibilities in the dissection of the protein(s) induced during the infection process. These proteins may further be correlated to the gene(s) and other related molecular tools that will help for the marker-assisted breeding and/or gene editing for this distressing disease among the major cereal crops.

## 1. Introduction

The fungal pathogen *Rhizoctonia solani* has pathogenicity over a broad range of host plants. *R. solani* causes sheath blight (ShB) in rice and is responsible for up to 25% of yield losses [1]. This necrotic fungal pathogen has a considerable effect on the production of an extensive range of plants imperative to humanity, including the 15 largest food crops worldwide, and causes root rot disease in wheat and barley [2]. Despite cost-effective treatments and the large impact of the pathogen, very little is known about how it causes disease in the cereal crops. ShB disease management has been largely achieved through systemic fungicides and certain non-systemic fungicides, namely, captaf, mancozeb, pencycuron [1-(4-chlorobenzyl)-1-cyclopentyl3-phenylurea], etc. However, these chemical fungicides harshly affect environmental conditions as pertains to beneficial microorganisms and others present in the atmosphere [3]. Uses of the chemical fungicides raise the cultivation outlay of the crops. Resistance to *R solani* varied among cereal crop species [4], thus, the information on resistant host cultivars and the defense mechanisms studied against ShB have become an essential concern in the present scenario. Several genetic loci have been identified that can confer minor resistance against *R. solani* [5]. Therefore, identification of tolerant or resistant genotypes is of prime importance for categorizing the crops for susceptibility and resistance to this necrotic disease [6].

Few studies identified that farm soil type can also influence the severity of the disease. There are several reports where resistance has been identified in the wild rice and other hosts against different diseases [7]; further, it was also reported that the wild relative of rice has not only an agronomically important gene, but also superior genetic diversity in resistant genes as compared to cultivated rice. In cereal crops, resistance against *R. solani is* governed by quantitative traits [8], and these traits might be associated with broad-spectrum resistance. Up to 50–60% of defense-related alleles are lost in cultivated rice in resemblance to wild rice [9]. The researchers’ intentions are to sort new rice genotypes with improved disease resistance and to improve yield potential using genomic and molecular information to shatter the rice yield plateau [10]. Few early studies attempted to look for the detail of the mode of rice infection by sheath blight pathogen, which can be further utilized for the conception of resistance mechanisms against *R. solani* [1]. *R. solani* isolates produce hydrolytic enzymes such as cellular cellulase, pectolytic, and protease enzymes which help in the invasion of the host’s primary line of defense [11]. These induced toxins produced by *R. solani* have corresponded with sheath susceptibility in rice. Several mechanisms, including redox regulation, reactive oxygen species (ROS) formation, signal transmission, and metabolic alterations, were involved in the defensive routes against the *R. solani* pathogen in cereals [12]. The biomolecules which were induced after *R. solani* infection have been linked to basal resistance in several crops including rice [13]. 

Rice resistance to ShB is a typical quantitative attribute governed by many genes or quantitative trait loci (QTLs) [14]. No rice variety has been reported to be completely resistant to *R. solani*, despite the discovery of more than 60 SB resistance quantitative traits (QTLs). Additionally, no SB resistance QTLs have been cloned [8]. Thus, there is a need to investigate the molecular as well as the proteomic approaches in the wide host range to confer resistance sources against the ShB pathogen. Researchers also provide proteomics-based tools, which serve as an effectual method for investigating the molecular responses of plants to various biotic stresses [15]. Lee et al. [16] recognized a precise 3-β HSD protein in resistant cultivated rice varieties. LSBR-5 connected with a response to infection by *R. solani* using 2-D and mass spectrometry (ESI Q-TOF MS). Similarly, a set of 319 differentially accumulated proteins (DAPs) after *R. solani* [13] was also studied for the investigation of resistance. A study of comparative metabolome and proteome profiles of the rice lines (wild-type and *AtNPR1*-transgenic) before and after *R. solani* infection [17] also revealed the host–pathogen interaction. 

*R. solani* infects a broad range of cereal crops (rice, wheat, maize, barley, etc.) and no proper evidence has been found on how this necrotic fungal pathogen infects the host. The comparative defense response of different hosts has also been lacking until now, and effort should be put into exploring molecular targets for sheath blight-resistance which can be used for introgression to the rice and other cereals. Therefore, differential proteomic analysis of infected hosts in cereals (i.e., rice, wild rice, barley) may play an important role in finding resistance to ShB. Thus, the present study was carried out in order to determine the proteomic and biochemical signals that occur during the cultivated rice–*R. solani*, wild rice–*R. solani*, and barley–*R. solani* interactions for the purpose of searching for resistance possibilities in cereal crops.

## 2. Materials and Methods

### 2.1. R. solani Isolate, Plant Materials and Inoculation Procedure

Seeds of cultivated rice cv. Pusa Basmati-1, wild rice accession *O. grandiglumis*, and barley cv. NDB-1445 were collected from N.B.P.G.R., New Delhi, I.R.R.I., Philippines, and the Department of Genetics and Plant Breeding, A.N.D. University of Agriculture and Technology, Kumarganj, Ayodhya, U.P., India, respectively. They were sterilized with 5% bleach for 10 min and thoroughly rinsed with distilled water. Germinated seeds were placed in petri dishes at 28 °C for three days under dark conditions and further grown in pots (45 cm × 60 cm) filled with autoclaved pot mixture (30 mg N, 9.7 mg P, and 18.5 mg K per kilogram of soil in the form of urea, single superphosphate, and muriate of potash, respectively). All host plants were grown in a greenhouse at 25 ± 3 °C (14 h light/10 h dark cycle). The *R. solani* strain D-14, belonging to the AG1-IA anastomosis group, was collected from the Rice Pathology Laboratory, G. B. Pant University of Agriculture and Technology, Pantnagar, India. It was grown on potato dextrose agar at 28 ± 1 °C for 6 days and utilized for the inoculation. Approximately 0.2 mg of a 4-day-old immature sclerotium of *R. solani* grown on a PDA *(potato dextrose agar)* medium was used as inoculum. The sclerotium includes agar plugs embedded beneath the six-week-old host plants. The inoculated host plants’ sheaths were covered with sterilized absorbent cotton to maintain humidity. The host plants without *R. solani* inoculation were maintained as healthy controls. The experimentation employed a completely randomized block design. Each treatment had three replications with three pots in each *replication,* and the experiments were repeated twice. After inoculation, all plants were transferred for 24 h. at 28 °C in complete darkness and up to 100% humidity, then transferred at 20 °C for 14 h in the light, and finally, for 10 h in the dark at 60% humidity. The relative lesion height (RLH = lesion height/plant height × 100) was recorded as described by Singh et al. [18]. The disease severity was calculated based on the RLH percentage. The data of RLH was converted into a disease index based on a disease score with a 0 to 9 rating scale. For the biochemical and proteomic analysis, leaf sheath tissue of ~3 cm was harvested above the 2 cm lesion with *R. solani* in the inoculated hosts, as described by Lee et al. [16], in order to avoid contamination. 

### 2.2. Estimation of Total Phenol Content, Peroxidase (PO) and β-1, 3-Glucanase

For biochemical assays, infected and mock leaf samples were collected from rice, wild rice, and barley at 24, 48, 72, and 96 hpi (hours post-infection). For phenol estimation in samples, 100 mg of freshly collected infected and control samples were ground in liquid N_2_. The homogenized sheath was collected in 10 mL of solvent (80% aqueous acetone) for 1 min, and transferred in test tubes and centrifuged for 15 min at 1000× *g* (REMI C-224 Centrifuge). After centrifugation, a clear supernatant was collected. The amount of total phenolics in the leaves in both the control and inoculated conditions were determined according to the Folin–Ciocalteu procedure [19]. Next, 2 mL samples in three replicates were transferred into test tubes; 1.0 mL of Folin–Ciocalteu’s reagent and 0.8 mL of sodium carbonate (7.5%) were added in each replicate. The tubes were assorted thoroughly and permitted to stand for 30 min. Absorption at 765 nm was measured (Systronics UV-vis spectrophotometer, Ahmedabad, Gujarat, India). The total phenolic content was stated as gallic acid equivalents (GAE) in milligrams per gram of the sample dry material. The phenol content of the extract was expressed as μg, and the phenol equivalent released g^−^^1^ of leaf tissue. For peroxidase activity assay, 100 mg of both freshly infected and control samples were ground in liquid N_2_. The ground powder was used for the estimation of peroxidase activity by adopting the method described by Srivastava [20] and scoring the absorbance at 420 nm. The enzyme activity in the tissue sample was represented by the ktkatg^−^^1^. Similarly, 100 mg of freshly ground tissue of infected and control samples were used to estimate β-1, 3-glucanase activity by the laminarin-dinitrosalicylic acid method [21]. The enzyme activity was expressed as μg, and glucose released min^−1^ mg^−1^ proteins. All of the biochemical analyses were repeated thrice in this experiment.

### 2.3. Protein Extraction for 2D Analysis

Leaf-sheaths from both inoculated and non-inoculated (control) plants of different hosts (rice, wild rice, and barley) were collected after 48 hpi. Fresh leaf samples (1 g of control and inoculated) were crushed into a fine powder using liquid nitrogen. The ground homogenate was extracted in 10% TCA in acetone containing 0.07% DTT and then kept at −20 °C for 1 h, followed by centrifugation for 15 min at 35,000× *g*. The pellets were centrifuged once for 15 min at 35,000× *g*, kept at 20 °C for an hour, and then given a second wash with ice-cold acetone containing 0.07% DTT. When the supernatant was clear, the washing procedure was repeated four or five times. The final precipitated pellets were lyophilized using the Freeze Dryer lyophilizer (Sew, New Delhi, India). Proteins from the leaves were extracted as described by Koller et al. [22]. The experiments were repeated thrice from the three replicate samples.

### 2.4. Isoelectric Focusing (IEF) and Polyacrylamide Gel Electrophoresis (PAGE)

A total of 10 mg of the dried powder (control and treated) was suspended in 350 μL of buffer, containing 7 M urea, 2 M thiourea, 4% CHAPS ((3-cholamidopropyl) (dimethylammonio)-1-propanesulfonate) detergent, 0.5% ampholytes (pH 3–10), and 0.7% DTT. The supernatant was distributed in 100 μL aliquots and kept at −80 °C for further use. In order to determine optimum 2-DE gel conditions, a broad range (pH 3–10) IPG strip for the first dimension and a 12% linear polyacrylamide gel for the second dimension were carried out. The preponderance of the protein spots was perceived in the center of the gel, thus, the 2-DE gel with a pH 4–7 range and a 12% linear polyacrylamide gel were used. A total of 500 µg of protein, assayed by the Bradford method [23], was loaded onto a 17 cm (pH range 4–7) linear IPG (immobilized pH gradient) strip (BIO-RAD, Hercules, CA, USA) into the rehydration/equilibration tray with slight modification (Kumar et al. [24]). The rehydration tray containing the IPG strips was covered by 2 to 3 mL of mineral oil to avoid evaporation during the rehydration process. The first-dimensional electrophoresis (isoelectric focusing) was carried out with a PROTEAN IEF Cell (BIO-RAD, Hercules, CA, USA) at 250 volts for 20 min in linear mode, followed by 8000 volts for 2.5 h in linear mode, followed by 8000 volts at ~30,000 volt-hour mode in rapid mode. The IPG strips were subjected to second-dimensional electrophoresis after 10 min in equilibration buffer 1 (6 M urea, 0.375 M tris, pH 8.8, 2% sodium dodecyl sulphate (SDS), 20% glycerol, and 2% (w/v) DTT), followed by 10 min in equilibration buffer 2 (6 M urea, 0.375 M tris, pH 8.8, 2% SDS, 20% glycerol and 2.5% iodoacetamide). The equilibrated IPG strips were washed in sterile, distilled water, and then placed on top of a 12% non-gradient and 18 cm × 20 cm polyacrylamide-bisacrylamide gel. No stacking gel was used [24]. Bromophenol blue-containing 1% agarose was used to seal the IPG strip. The following operating conditions were used for the second electrophoresis: 30 min of constant 16 mA at 6 °C, followed by constant 30 mA in a vertical electrophoretic dual gel unit PROTEAN II XI (BIO-RAD, Hercules, CA, USA). Coomassie Brilliant Blue R250 was used to stain the gels of all the tested samples. 

### 2.5. Image Acquisition, Spot Digestion and Identification

Images of the stained gels were obtained by Chemidoc (BIO-RAD, Segrate (Milan), Italy) using Quantity One software version 4.6.3 (BIO-RAD, Hercules, CA, USA). Spot detection and matching analyses were conducted first with the PDQuest Advanced software (version 8.0) (Bio-Rad, Hercules, CA, USA), and then manually checked a second time. Gel images were analyzed using stepwise spot detection and spot matching followed by differential expression analysis. Protein spots among the gels showed increased/decreased intensities with more than 1.5-fold selected for identification. PDQuest employs a Student t-test and enumerates spots with a differential intensity of significant levels. Differentially expressed protein spots of interest were excised from gels using the spot picker ‘Investigator ProPic’ (Genomic Solutions, Huntingdon, UK) and collected in 96-well PCR plates. The digestion of protein and spotting of peptides on the MALDI-TOF target plate was carried out using the protein digester ‘Investigator ProPrep’ (Genomic Solutions, Huntingdon, UK) as described by Shevchenko et al. [25] with slight modifications. The gel plugs were de-stained and dehydrated by washing three times (~10 min) with 25 mM NH_4_HCO_3_-50% acetonitrile (ACN) (1:1). Dried gel plugs were treated with freshly prepared 10 mM DTT in 50 mM NH_4_HCO_3_ for 45 min at 56 °C. After incubation, the DTT was replaced quickly by 55 mM iodoacetamide (freshly prepared) at the same volume in 50 mM NH_4_HCO_3_ for 30 min and, finally, dehydrated with 100% ACN. The dried gel pieces were incubated for 12 h at 37 °C with 25 mM NH_4_HCO_3_ containing 0.02 μg/μL of mass spectrometry grade trypsin (Promega, Madison, WI, USA). The resulting peptides were extracted twice from the gel pieces, using a peptide extraction buffer.

The digested protein samples were analyzed by mass spectrometry in the same manner that was described earlier [25]. According to the instructions provided by the manufacturer, digested protein samples were desalted and concentrated using C-18 ZipTips (Millipore, Billerica, MA, USA). The ZipTips were eluted from the MTP 384 target plate using 2 L of a saturated solution of a-cyano-4-hydroxycinnamic acid (HCCA) (Sigma-Aldrich, St. Louis, MO, USA) dissolved in 50% aqueous cyanohydrin and 0.2% trifluoroacetic acid. The Autoflex II TOF/TOF 50 mass spectrometer from Bruker Daltonik GmbH in Leipzig, Germany was used to acquire mass spectra of digested proteins. The instrument was set to the positive reflectron mode, and the detection range was 500–3000 m/z. Before the acquisition, an external calibration to a spectrum was carried out, with the spectrum being acquired for a mixture of peptides whose masses ranged from 1046 to 2465 Da. After that, the proteolytic masses which were obtained were put through the Flex Analysis v.2.4 program so that peak detection could take place. In order to identify the proteins contained in the rice database (*O. sativa* and other green plant data, EMBL/GenBank/DDBJ), peak list submissions were made to the UniProtKB and Swiss-Prot databases through the use of the Mascot search engine found at http://www.matrixscience.com (accessed on 9 and 27 November 2019).

### 2.6. Protein Analysis and Chromosome Localization of Differentially Expressed Protein

Biochemical characteristics such as molecular weight (M.Wt), isoelectric focusing (PI), and signal peptides were examined using the SIB Bioinformatics portal (http://www.expasy.org, accessed on 27 November, 2019). Cellular/subcellular targeting sites were predicted using WoLF-PSORT (https://wolfpsort.hgc.jp/ accessed on 27 November 2019), Predator (link), and TargetP1 servers (http://www.cbs.dtu.dk/services/TargetP-1.1/index.php accessed on 27 November 2019). Physical locations of genes were obtained from the Gramene (http://www.gramene.org/ accessed on 27 November 2019) and Phytozome databases (https://phytozome.jgi.doe.gov/pz/portal.html/ accessed on 27 November 2019) and finally represented using the GGT 2.0 version tool. Bioinformatics approaches such as finding out the ORFs of genes as well as their expressional graphs were studied, and data were obtained from NCBI’s (National Centre for Biotechnology Information) Gene Database (https://www.ncbi.nlm.nih.gov/gene, accessed on 27 November 2019) and the ORF Finder of NCBI (https://www.ncbi.nlm.nih.gov/orffinder, accessed on 27 November 2019). 

### 2.7. Co-Expression Network and Gene Ontology Analysis

For the purpose of mapping, the co-expression network of differentially expressed proteins in cultivated rice cv. Pusa Basmati-1, wild rice accession *O. grandiglumis,* and barley cv. NDB-1445 under *R. solani* inoculation were considered individually as input in the STRING 11.0 version (https://string-db.org/ accessed on 27 November 2019) and repeated twice [26]. For this, protein ids/protein sequences were used to obtain the co-expressing primary interactors, considering the maximum number of primary interactors was set to 500 and the interaction score cut off value was set at ≥0.700. Then, a complete list of primary interactors and differentially expressed proteins was used as input to construct the co-expression network and gain insight into a functional classification for *R. solani*-induced proteins in cultivated rice cv. Pusa Basmati-1, wild rice accession *O. grandiglumis,* and barley cv. NDB-1445. For enrichment analysis, gene ontology was performed using STRING 11.0 to obtain insight into the enriched category (FDR corrected *p*-value ≤ 0.05). 

### 2.8. Data Analysis

The SAS (Statistical Analysis Systems) software (version 7.0, SAS Institute, Cary, NC, USA) [27] was utilized for the data analysis in the experiment, which was laid out in a complete randomized design (CRD) with 3 replications. In the disease scoring experiment, cultivated rice cv. Pusa Basmati-1 was used as a susceptible check and wild rice accession *O. grandiglumis* was used as a resistance check. The protein spots in the gels were captured by PDQuest software, which employs a Student t-test analysis.

## 3. Results

The *Oryza* spp. (cultivated rice cv. Pusa Basmati-1 and wild rice accession *O. grandiglumis*) and barley cv. NDB-1445 ranged from very susceptible to mildly resistant to sheath blight. The cultivated rice cv. Pusa Basmati-1 showed the highest visual rating in terms of infection under field and the detached leaf method, but *O. grandiglumis* was found to have the lowest infection across all tested hosts. In field conditions, the average lesion number was also largest in Pusa Basmati-1, but it was lower in *O. grandiglumis* than in the barley cv. NDB-1445 and the other tested hosts (Figure 1a–c).

### 3.1. Biochemical Assay

Biochemical assays of total phenol, peroxidase, and β-1,3-glucanase were performed in order to assess the *R. solani* infection in cultivated rice cv. Pusa Basmati-1, wild rice accession *O. grandiglumis,* and barley cv. NDB-1445 on different time intervals, such as at 24 hpi, 48 hpi, 72 hpi, 96 hpi, 120 hpi, separately and in control condition. The hosts’ plants show variation in phenol, peroxidase, and β-1,3-glucanase accumulation at different time intervals on *R. solani* inoculation (Figure 2a–c). The accumulation of total phenol content increased after inoculation with *R. solani* in Pusa Basmati-1, *O. grandiglumis*, and NDB-1445 as compared to the control condition (Figure 2a,b). Accumulation of total phenol was much higher and appeared earlier in *O. grandiglumis* compared to other hosts, and further increased up to 120 hpi in both rice (cv. Pusa Basmati-1 and barley cv. NDB-14445). The total fold change in phenol content ranged from 0.94-fold (Pusa Basmati-1 at 120 hpi) to 2.13-fold (NDB1445 at 7 2 hpi). The time course of change in peroxidase among inoculated cultivated rice cv. Pusa Basmati-1, wild rice accession *O. grandiglumis,* and barley cv. NDB-1445 appeared to change from >1 to 1.6-fold. Activity of peroxidase was started up at minimum in Pusa Basmati-1 and *O. grandiglumis* at 24 hpi, and at maximum in barley cv. NDB-1445, at 72 hpi and 96 hpi. Similarly, the activity of β-1,3-glucanase in wild rice *O. grandiglumis* was the highest among all of the hosts and increased consistently after 24 hpi up to 120 hpi. β-1,3-glucanase activity ranged from a 1.2-fold (cultivated rice cv. Pusa Basmati-1 and barley cv. NDB-1445 at 24 hpi) to a 2.6-fold change (*O. grandiglumis* at 96 h). 

### 3.2. R. solani Induced Proteome Shift in Cultivated Rice, Wild Rice, and Barley

In the 2-DE maps, more than 1840 protein spots and an average of 613 protein spots were viewed with Coomassie brilliant blue staining across all the hosts. A total of 672 spots were identified, of which 605 were matched among the gels. In a comparison analysis between inoculated and non-inoculated rice cv. Pusa Basmati-1, 67 spots were differentially expressed on *R. solani* inoculated gel. After 48 hpi of inoculation, 15 distinct protein spots were found with significant differential accumulation in response to pathogen infection. Spots were considered to be differentially accumulated when a fold change of 2 in their relative volume was observed on all three experimental repeats (Figure 3a,b; Table 1). A total of eight up-regulated and four down-regulated protein spots were selected based on high percent reduction and induction of the protein for the MALDI-TOF analysis (Figure 3c,d). The up-regulated proteins were identified as ribulose bisphosphate carboxylase large chain, thioredoxin peroxidase, alcohol dehydrogenase, fructose-1, 6-bisphosphatase, O-methylthiopentene, OsClp1-putative protease homologue, thioredoxin, and hypothetical protein OsJNBa0069E14). The four down-regulated proteins were identified as magnesium-chelatase subunit chlI, alcohol dehydrogenase, catalase domain-containing protein, and ribulose bisphosphate carboxylase large chain precursor (Table 1). Induced protein location and their respective ORFs are mentioned in Appendix A.

In the comparative analysis of wild rice *O. grandiglumis**,* 559 spots were detected, out of which 515 spots were matched to all the gels and 44 protein spots were perceived to have a different relative expression. Among 44 protein spots, 32 spots were up-regulated and 12 protein spots were down-regulated. The comparative protein profiles of wild rice accession *O. grandiglumis* showed that 14 protein spots were differentially accumulated between the control and *R. solani* treated samples (Figure 4a,b). Finally, nine up-regulated and three down-regulated protein spots were selected for the MALDI-TOF analysis. In the up-regulated spots, putative cysteine proteinase inhibitor, proteinase inhibitor type-2, kinesin motor domain-containing protein, chlorophyll a-b binding protein, kinesin motor domain-containing protein, ZOS5-12 C_2_H_2_ zinc finger protein, acyl carrier protein desaturase, and down-regulated speckle-type POZ protein were identified (Figure 4c,d; Table 2). Induced genes and their respective ORFs are mentioned in Appendix A.

In barley cv. NDB-1445, comparative analysis revealed 609 protein spots, of which 570 spots were matched to all of the treatment gels, and 39 spots which significantly varied in their expressions. Among the 39 spots detected, 30 spots were up-regulated and 9 spots were down-regulated (Figure 5a,b). Further protein spots showing more than twofold increased intensities were further processed for identification purposes. The up-regulated spots were identified as a heat shock factor, peptidase ClA papain, PhotosystemII PbsO, and ribonucleoside-diphosphate reductase small chain. Down-regulated spots were identified as chlorophyll a-b binding protein 3C and a hypothetical protein (Figure 5c,d, Table 3). Induced genes and their respective ORFs are mentioned in Appendix A.

### 3.3. Chromosome Location of Differentially Expressed Protein on the Chromosomes

Genes encoding fifteen of the differentially expressed proteins from Pusa Basmati-1, nine proteins from wild rice accession *O. grandiglumis* (Appendix A), and six from barley cv. NDB-1445 (Appendix A) were located on their respective chromosomes. Four proteins of cultivated rice and one from wild rice were found to be located on rice chromosome 3, followed by two proteins from cultivated and one from wild rice which were located on chromosome 10, two proteins located on chromosome 7 and chromosome 12, and one each on chromosomes 2, 4, 5, 6, 8, and 11. Additionally, genes encoding the highest no. of differentially expressed proteins were assigned to chromosome 3. In barley cv. NDB-1445, the maximum no. of differentially expressed proteins was found on chromosome 7, followed by chromosome 2, whereas one differentially expressed protein was seen on chromosome 5 (Appendix A).

### 3.4. Co-Expression Networks and Gene Ontology

In order to understand the biological significance of differentially expressed proteins upon *R. solani* infection, identified proteins were screened for their co-expression network and gene ontology using the STRING 11.0 version. The co-expression network was mapped using differentially expressed proteins along with their primary interactors. Primary interactors were extracted, and the co-expression interaction network was individually mapped for differentially expressed proteins of Pusa Basmati-1 (Figure 6a), *O. grandiglumis* (Figure 7a), and barley cv. NDB-1445 (Figure 8a). The gene ontology enrichment analysis of Pusa Basmati-1 identified that the differential proteins are majorly localized in the cytoplasm (GO:0005737), intracellular membrane-bounded organelle (GO:0043231), and chloroplast (GO:0009507). The significantly enriched biological process was related to the oxidation–reduction process (GO:0055114) and photosynthesis (GO:0015979). The result of significantly overrepresented terms for molecular functions lies majorly in catalytic activity (GO: 0003824) and oxidoreductase activity terms (GO:0016491) (Figure 6b, Appendix A). A similar analysis of differentially expressed proteins in *O. grandiglumis* showed a highly connected co-expression network, suggesting their association with biological function (Figure 7b). Gene ontology enrichment identified that differentially expressed proteins and primary interactors were majorly localized in the intracellular part (GO:0044424) and in the cytoplasm (GO:0005737). Similarly, an enriched biological process lies in the nitrogen compound metabolic process (GO:0006807), gene expression (GO:0010467), and translation (GO:0006412) terms. The molecular function category showed enriched terms such as organic cyclic compound binding (GO:0097159), nucleic acid binding (GO:0003676), and chlorophyll binding (GO:0016168) (Figure 7b, Appendix A). Using a similar strategy, differentially expressed proteins from barley were also screened for primary interactors, co-expression network analysis, and GO enrichment analysis (Figure 8a). The significantly enriched cellular components belong to the chloroplast (GO:0009507) and the photosynthetic membrane (GO:0034357), whereas the significantly enriched biological process involves photosynthesis (GO:0015979) as well as the generation of precursor metabolites and energy (GO:0006091). The enriched molecular function GO terms for the co-expression network for barley involve metal ion binding (GO:0046872) and cofactor binding (GO:0048037) (Figure 8b, Appendix A). Intriguingly, *R. solani* induced differentially expressed proteins, co-expressed with cellular proteins that majorly determine photosynthesis and primary metabolic processes, showing enrichment of common GO terms. A pathway analysis of biological processes and protein class was conducted, in which differently abundant proteins after *R. solani* inoculation with the hosts were involved, as depicted in Figure 9a,b. As a whole, seven functional groups based on molecular process are most clearly affected by the *R*. *solani* infection, including protein binding (03), enzyme regulator activity (03), transferase activity (03), lyasese activity (04), hydrolase activity (07), and protein binding (18). With regard to the cellular components’ function, the key groups containing intracellular (32), cell periphery (13), plasma membrane (10), protein complex (8), endomembrane system (7), intracellular organelle part (5), external encapsulating structure (4), non-membrane bounded organelle (4), and organelle lumen (3) were found (Figure 9b). The present investigation suggests that *R. solani* inoculation in cereal crops induced conserved signals which affect vital processes of the inoculated hosts’ growth and survival.

## 4. Discussion

Application of fungicide in rice cultivation is a common practice, and if it is not applied precisely, it could adversely impact the environment as well as human beings and other living organisms [28]. Studies on the rice sheath blight resistance mechanism are limited due to the lack of resistant donors in cultivated cultivars [29]. So far, interaction between the fungus and the plant has been studied [30], but there have been only a few reports on *R. solani* interactions with a wide range of hosts. Thus, for the identification of the differentially expressed biochemical signals and proteins among the three hosts of *R. Solani*, a comparative proteomics study has been performed. Similarly, studies on ShB resistance genes in rice and their processes have primarily utilized cultivated rice as a host for a long time [31]. However, due to the polygenic nature of ShB resistance, no significant ShB resistance genes or rice cultivars demonstrating complete resistance to *R. solani* have been reported [32]. Thus, in the present study, three hosts (namely, cultivated rice cv. Pusa Basmati-1, wild rice accession *O. grandiglumis,* and barley cv. NDB-1445) have been selected for the resistance response against the *R. solani* pathogen. 

Recently, significant developments in proteomics have made substantial progress in understanding the physiological process governing rice resistance, associated signaling networks, and their function in triggering defense responses [16]. For the assessment of the biochemical response, we performed a comprehensive biochemical (total phenol content, peroxidase activities, and β-1, 3-glucanase), and proteomic analysis of the different hosts (cultivated rice cv. Pusa Basmati 1, wild rice accession *O. grandiglumis,* and barley cv. NDB-1445) after *R. solani* infection in order to comprehend how host–pathogen interactions work in rice sheath blight resistance. The host plants show variation in phenol, peroxidase, and β-1,3-glucanase accumulation at different time intervals after *R. solani* inoculation [33]. From the biochemical analysis, it is clear that accumulation of phenol, peroxidase, and β-1,3-glucanase was generally higher in infected samples as compared to the control. With regard to pathogen attacks, plants generally produce and accumulate defense-related compounds such as phenols, peroxidases, β-1,3-glucanases, chitinases, etc. Phenol and peroxidase activities in wild rice accession and barley cv. NDB-1445 showed significant induction after *R. solani* inoculation. Both enzymes showed maximum activity at 72 hpi in wild rice accession, compared with susceptible cultivated rice cv. Pusa Basmati-1. Plant phenolic compounds work as a structural barrier against pathogen attacks [34]. Plant peroxidases play an essential role in catalyzing the oxidation of different reductants in cells. These enzymes are active regulators of auxin metabolism, lignin and suberin synthesis, cell wall component cross-linking, phytoalexin synthesis, ROS (reactive oxygen species) metabolism, and RNS reactive nitrogen species), and are involved in disease resistance in plants [35]. The activities of these enzymes were found to be altered after fungal infection [35]. According to the earlier reports, the activity of peroxidase was significantly enhanced in rice leaves after artificial inoculation with *M. oryzae* and *R. solani* [36]. A similar report of high concentrations of defense enzymes such as 1,3-glucanase, phenylalanine ammonia lyase (PAL), peroxidase (POX), polyphenol oxidase (PPO), lipoxygenase (LOX), and defense protein hydroxyproline-rich glycoproteins (HRGPs) positively correlated with increased downy mildew resistance in pearl millet (*Pennisetum glaucum* (L.) after *Sclerospora graminicola* (Sacc.) inoculation [37]. These results are consistent with our current inquiry.

Similarly, the induction of β-1, 3-glucanase is highest at 72 hpi in the wild rice accession *O. grandiglumis,* and increased consistently up to 120 hpi in comparison to other hosts. The β-1,3-glucanase enzyme plays a crucial part in the breakdown of fungal cell walls by hydrolyzing chitin, where it has been demonstrated that cell wall fragments cause plants to mount a defense response [37]. Similarly, a study showed that β-1,3-glucanase enzyme activity significantly increased following a pathogen challenge in Taipei (TP) 309 containing the p*i54* gene at 120 hpi, but were significantly lower in other rice genotypes [38]. 

In cultivated rice cv. Pusa Basmati-1, upon *R solani* infection, upregulation of eight proteins was identified: ribulose bisphosphate carboxylase large chain, thioredoxin peroxidase, alcohol dehydrogenase, fructose-1,6- bisphosphatase, O-methylthiopentene, OsClp1-putative protease homologue, thioredoxin, and the hypothetical protein OsJNBa0069E14. In previous studies, it has been reported that Rubisco (ribulose bisphosphate carboxylase) is inactivated by ROS after biotic stress and is mostly downregulated [17,39]. However, Wu et al. [40] reported that the sugarcane mosaic virus (SCMV) infection affects photosynthesis-related proteins, which were down-regulated in maize (*Zea mays*) seedlings, with the exception of the Rubisco large subunit as well as the ferredoxin-NADP reductase and its isoforms. Induction of the ribulose bisphosphate carboxylase large chain precursor was also observed, and similar observations have been reported with abiotic and biotic stresses in rice. Similarly, the upregulation of Thioredoxin peroxidase might be due to its ability to mitigate oxidative stress along with thioredoxin reductase [13,41]. 

The photosynthetic protein ribulose bisphosphate carboxylase large chain was up-regulated, and the role of Rubisco is well studied in photorespiratory carbon oxidation and photosynthetic CO_2_ assimilation. It has also been reported that after biotic stress, Rubisco is destroyed during senescence and oxidative stressors, and inactivated by ROS. Induction of another protein, namely, thioredoxin peroxidase, again showed the defense mechanism in rice after *R. solani* infection [13]. Thioredoxin reductase and thioredoxin peroxidase work together to mitigate oxidative stress [13,41]. Generally, fungal infections create hypoxia/anoxia conditions in their host plant according to which plants adapt mechanisms to compensate for the energy crisis. Alcohol dehydrogenase (ADH) is an important component in the maintenance of a primary energy metabolism in higher plants, and seems to be engaged in aerobic metabolism under specific stress situations [42] such as low temperature, water stress, or ozone exposure. In several studies, alcohol dehydrogenase (ADH) and pyruvate carboxylase are central enzymes in fermentative metabolism, and were reported after biotic stress [43]. Inoculation of *R. solani* to the rice cv. Pusa Basmati-1 induces the accumulation of O-methyltransferases (OMT). Plant O-methyltransferases (OMT) play an important role in lignin biosynthesis and stress tolerance in plants [44]. A greater accumulation of OMT was also observed in barley after infection with the fungus *Blumeria graminis* [45].

After *R. solani* inoculation, rice cv. Pusa Basmati-1 also showed down-regulation in proteins, namely, magnesium-chelatase subunit chlI, alcohol dehydrogenase, catalase domain-containing protein, and ribulose bisphosphate carboxylase large chain precursor. An increased relative abundance of anaerobic metabolisms under salt stress was reported by the increased relative abundance of FBP aldolase, other glycolytic enzymes, and enzymes involved in ethanolic fermentation and glycolate metabolism in rice seedlings. After inoculation with *R. solani* OsClp10, a putative Clp protease homolog protein showed down-regulation. The important function of the chloroplast Clp protease has been proven in earlier research employing several transgenic plants. Plants with lower levels of the plastomic ClpP1 or other Clp proteolytic core elements have a reduced ability to differentiate their chloroplasts, generate shoots, and maintain overall plant viability. The Rubisco small chain protein ribulose bisphosphate carboxylase also showed down-regulation after inoculation with *R. solani.* It is well known that infected plant cells have lower levels of the Rubisco protein, a vital photosynthesis enzyme, as a result of pathogens attacking chloroplasts and causing their destruction [46].

Wild rice accession *O. grandiglumis* showed induction of a cysteine-type endopeptidase inhibitor. The induced proteinase inhibitors (PIs) are categorized under the PR6 family, which plays an important role in plant defense. Additionally, it has been found that plant PIs influence plant immunity by inhibiting pathogen proteases or controlling endogenous plant proteases [47]. The majority of cysteines found in plants are found in tiny proteins that act as inhibitors of the C1A papain-like family of cysteine proteases (CysProt). It has been assumed that PhyCys has a defense function against viruses and pests based on its up-regulation in response to biotic, stress-related signals. It has been demonstrated that the *A. thaliana* serine protease inhibitor (UPI) aids in defense against the necrotrophic fungi *Alternaria brassicicola* and *Botrytis cinerea* [48]. Recently, Zhang et al. [49] also reported a novel host defense mechanism in rice, where an NLR receptor protects a host protease inhibitor that is targeted by rice fungus *Magnaporthe oryzae* and promotes resistance against it.

The chlorophyll *a*/*b*-binding protein (CAB), related to metabolism and energy, showed up-regulation after *R. solani* inoculation at 48 hpi by the cultivated rice cv. Pusa Basmati-1 and wild rice accession *O. grandiglumis.* The CAB protein has a role in regulating the expression of sugar-related genes in plants. Following pathogen infection, several genes for chlorophyll a/b binding proteins were down-regulated, slowing photosynthesis and limiting the availability of food resources such as sugar [50]. The transcription of photosynthetic genes, including ribulose 1,5-diphosphate carboxylase and genes that bind chlorophyll a/b, was also found to be inhibited in an interaction between rice and *R. solani*, in both the susceptible and resistant interactions. However, wild rice accession *O. grandiglumis* induced the chlorophyll *a*/*b*-binding protein, probably due to the availability of food under stress.

The kinesin motor domain-containing protein was up-regulated after *R. solani* inoculation. It has been reported that plant stress tolerance depends on many factors, including signals generated by mitogen-activated protein-kinase modules, which play a crucial role under stress. After inoculation with *R. solani,* wild rice accession *O. grandiglumis* showed a high level of expression of C_2_H_2_ zinc finger (C_2_H_2_-ZF) proteins. Zinc finger transcription factors (TFs) had already been discovered in other plants, and it has been reported that these TFs contribute to vital biological functions during vegetative growth, reproductive development, and stress responses [13]. Several studies identified that a CCCH-type zinc finger protein regulates the interaction of pathogens in cotton as well as in rice [51]. The zinc finger protein (ZFP) (N2), which is associated with defense-related transcriptional factors, was induced in a MoHrip2 elicitor and treated plants as compared to controls. Similarly, acyl carrier protein desaturases and speckle-type POZ showed down-regulation after *R. solani* inoculation in *O. grandiglumis.* The vital roles that stearoyl-acyl carrier protein desaturases play in UFA production in response to biotic and abiotic stressors are well documented [52]. 

The barley cv. NDB-1445 showed up-regulation of the heat shock transcription factor after *R. solani* inoculation. There is evidence to support the importance of heat shock factors (Hsfs) in stress sensing and signaling of various environmental stresses. Induced reactive oxygen species (ROS) in plants, particularly H_2_O_2_, are important components in biotic and abiotic stress responses and signaling mechanisms [53]. The heat shock protein-encoding gene OsHsf23 was induced after interaction with *R. solani* via activation of OsHsf23. Similarly, Bechtold et al. [54] also confirmed HSFA1b’s additional function in sustaining baseline disease resistance, which was not dependent on stress hormones, but did include H_2_O_2_ signaling. 

A protease (namely, peptidase CIA (MLOC_751664) papain) was also induced after *R. solani* inoculation by barley cv. NDB-1445. Proteases also appear to play key roles in plant pathogenesis [47,55]. With the genetic identification of RCR3, a secreted papain-like protease (C1) of tomato, the function of papain-like proteases has caused interest. The function of the resistance gene Cf-2, which mediates recognition of the Avr2 avirulence gene of the fungus *Cladosporium fulvum*, depends on RCR3 [55]. The role of cysteine proteinases from different plants has also been associated with disease resistance [47]. 

The *R. solani* interaction with barley cv. NDB-1445 induced protein photosystem II PsbO. Through the creation of ROS, which not only harms the elements of the photosynthetic electron transfer chain but also serves as an essential retrograde signaling molecule, photosystem II (PSII) plays a significant role in plant immunology. An interaction between tobacco leaf PSII repair and the control of cell death was also reported after infections by tobacco mosaic virus had been established in tobacco plants [56].

Proteins (namely, ribonucleotide reductase small subunit and the CAB protein) showed reduced expression after *R. solani* inoculation. In yeast, humans, and, probably, higher plants, ribonucleotide reductase (RNR) is known to be a significant target of DNA damage checkpoint mechanisms [57]. There is also a report that upon *Septoria tritici* (filamentous fungus) infection, a large number of proteins, including chloroplast CAB proteins, ribulose bisphosphate carboxylase, fructose-bisphosphate aldolase, etc., show a decrease in abundance and/or changed in phosphorylation status in wheat cultivars from 3 to 11 dpi. This finding suggests suppression of photosynthesis, changes in sugar metabolism, and an increase in sugar content [58]. It has been reported that high sugar levels decrease photosynthetic gene expression and increase mobilization and sugar transport [58], and that fungal diseases can target sugar transporters and activate the expression of the relevant genes for nutritional gain [59]. According to the findings of this research, the reaction of cultivated rice and other hosts (wild rice and barley) to *R. solani* can be explained in terms of the physiological state of the infected cultivated rice and the energy needed to induce a defense, which is supplied by proteins involved in energy metabolism (GAPDH, Rubisco). On the other hand, wild rice and barley were inducted into the defense modes of other, different proteins. A number of photosynthesis-related proteins underwent changes, indicating the dynamic impact of SA on photosynthesis. A decrease in the photosynthetic rate has occurred after treatment, as evidenced by the down-regulation of proteins such as Rubisco large subunit, phosphoenolpyruvate carboxylase, ADP-glucose pyrophosphorylase large subunit, and Rubisco small subunit in control condition only.

Results analyzed from the present study conclude that stress-related proteins, including defense, carbon metabolism, and energy synthesis, have been significantly affected in the selected cereal crops after *R. solani* infection, i.e., in cultivated rice cv. Pusa Basmati-1, and together dictate transition to a ‘defense mode.’ Some common stress response systems appeared to be activated in both resistant and susceptible lines, but the resistant wild rice accession *O. grandiglumis* produced significantly more defense-related proteins than barley cv. NDB-1445. From the different protein expression patterns detected in this study, it is hypothesized that the resistance of wild rice accession *O. grandiglumis* resulted from its ability to produce more defense proteins than the more susceptible rice and barley cv. NDB-1445. This ability was, possibly, a loss in the cultivated rice variety.

## 5. Conclusions

The present study reveals a complex response of *R. solani interaction in* three cereal hosts showing simultaneous induction of key proteins, their biochemical profile, and chromosomal and physiological mapping. Here, we propose that defense pathways involved in *R. solani* interaction are due to some key regulated pathways at the translational levels, such as pathogenesis-related protein, resistance gene/protein domains, signal transduction, proteins involved in energy metabolism, and protein synthesis, which were induced in different hosts. In field conditions, wild relatives of rice accession *O. grandiglumis* showed high resistance as compared to barley cv. NDB-1445 (moderately resistant) and cultivated rice cv. Pusa Basmati-1 (susceptible). The identified sources could be dissected and engineered in order to develop novel sheath blight resistance rice cultivars through breeding programs.

## Figures and Tables

**Figure 1 bioengineering-09-00589-f001:**
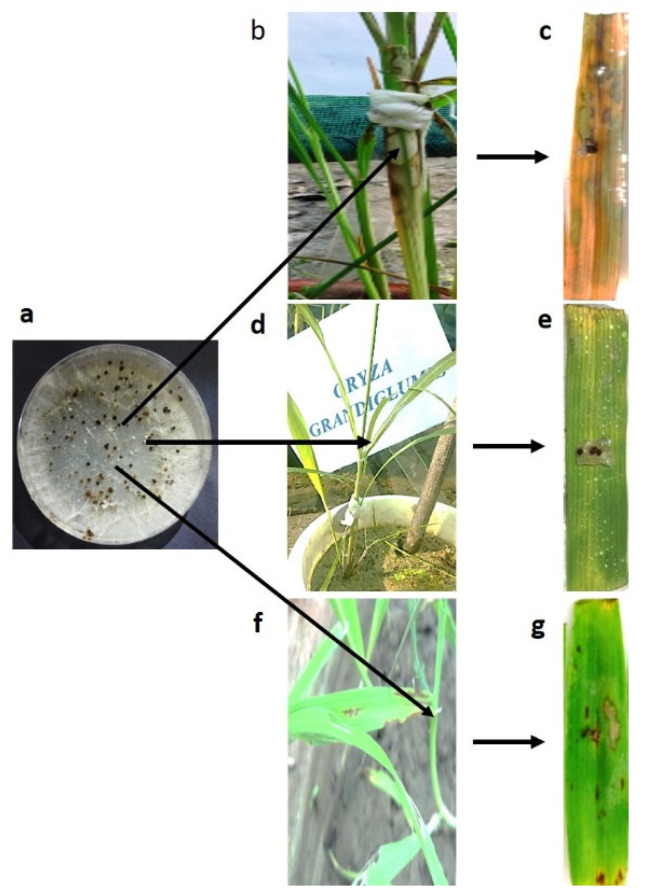
(**a**) *R. solani* (D-14 belonging to AG1-IA) isolate for the inoculation purpose to the hosts. (**b**) Disease reaction of *R. solani* on cultivated rice cv. Pusa Basmati-1. (**c**) Disease reaction of *R. solani* on cultivated rice cv. Pusa Basmati-1 by leaf detached method. (**d**) Disease reaction of *R. solani* on wild rice accession *O. grandiglumis*. (**e**) Disease reaction of *R. solani* on wild rice accession *O. grandiglumis* by leaf detached method. (**f**). Disease reaction of *R. solani* on barley cv. NDB-1445. (**g**) Disease reaction of *R. solani* on barley cv. NDB-1445 by leaf detached method.

**Figure 2 bioengineering-09-00589-f002:**
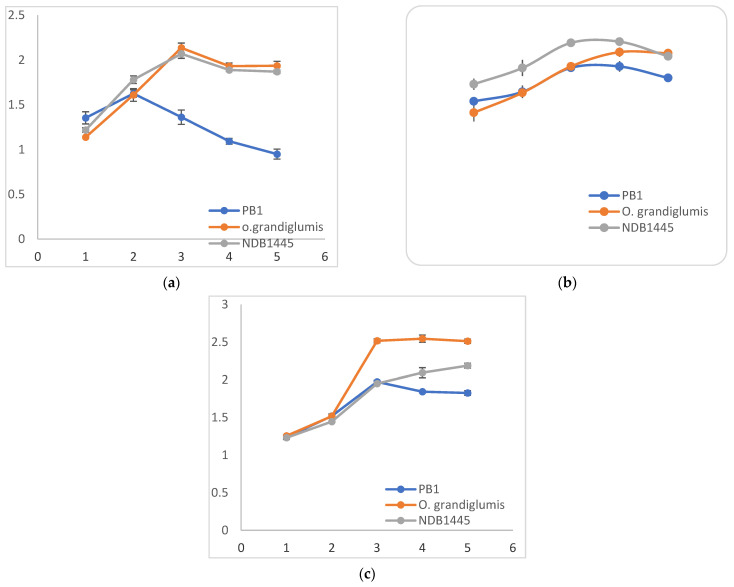
Enzymatic analysis of total phenol content, peroxidase (PO), and β-1,3-glucanase in cultivated rice cv. Pusa Basmati-1, wild rice accession *O. grandiglumis,* and barley cv. NDB-1445 in control and *R. solani* infected samples at different time intervals (24 hpi [1], 48 hpi [2], 72 hpi [3], 96 hpi [4], and 120 hpi [5]). (**a**) Fold change in enzymatic activity of total phenol content. (**b**) Fold change in enzymatic activity of total peroxidase content. (**c**) Fold change in enzymatic activity of total β-1,3-glucanase content. Mean differences were significant at ≤0.05% level.

**Figure 3 bioengineering-09-00589-f003:**
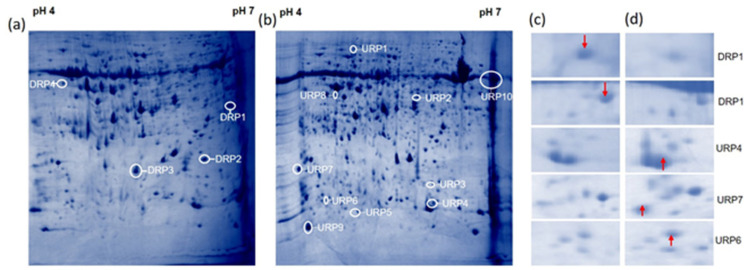
*R. solani*-induced proteome shift in cultivated rice cv. Pusa Basmati-1: Total proteins were extracted from cultivated rice cv. Pusa Basmati-1 and analyzed by 2-DE protein electrophoresis. In panel (**a**): complete 2D protein profile of cultivated rice cv. Pusa Basmati1 without *R. solani* infection. In panel (**b**): complete 2D protein profile of cultivated Pusa Basmati-1 48 h post-inoculation of *R. solani*. The spots which were more intense in the control are shown with a downward arrow (↓), and the spots which were more intense in the stressed cells are shown with an upward arrow (↑). After image analysis, 67 spots were found to be differentially expressed quantitatively between the two situations. In panels (**c**,**d**) are some of the representative differentially expressed enlarged regions of the protein profile from panels (**a**,**b**). Here, URP and DRP represent the up-regulated and down-regulated protein spots, respectively.

**Figure 4 bioengineering-09-00589-f004:**
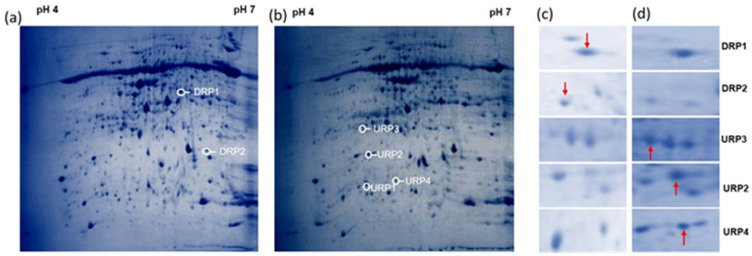
*R. solani*-induced proteome shift in wild rice accession *O. grandiglumis*: total proteins were extracted from wild rice accession *O. grandiglumis* and analyzed by 2-DE protein electrophoresis. In panel (**a**): complete 2D protein profile of wild rice *O. grandiglumis* without *R. solani* infection. In panel (**b**): complete 2D protein profile of wild rice accession *O. grandiglumis* 48 h post-inoculation of *R. solani*. The spots which were more intense in the control are shown with a downward arrow (↓), and the spots which were more intense in the stressed cells are shown with an upward arrow (↑). After image analysis, 44 protein spots were found to be differentially expressed quantitatively between the two situations. In panels (**c**,**d**): some of the representative differentially expressed enlarged regions of the protein profile from panels (**a**,**b**). Here, URP and DRP represent the up-regulated and down-regulated protein spots, respectively.

**Figure 5 bioengineering-09-00589-f005:**
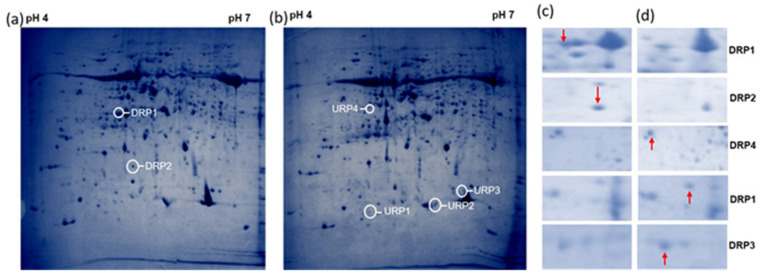
*R. solani*-induced proteome shift in barley cv. NDB-1445: total proteins were extracted from barley cv. NDB1445 and analyzed by 2-DE protein electrophoresis. In panel (**a**): complete 2D protein profile of barley var. NDB-1445 without *R. solani* infection. In panel (**b**): complete 2D protein profile of barley cv. NDB-1445 48 h post-inoculation of *R. solani*. The spots which were more intense in the control are shown with a downward arrow (↓), and the spots which were more intense in the *R. solani* cells are shown with an upward arrow (↑). After image analysis, 39 protein spots were found to be differentially expressed quantitatively between the two situations. In panels (**c**,**d**): some of the representative differentially expressed enlarged regions of the protein profile from panels (**a**,**b**). Here, URP and DRP represent the up-regulated and down-regulated protein spots, respectively.

**Figure 6 bioengineering-09-00589-f006:**
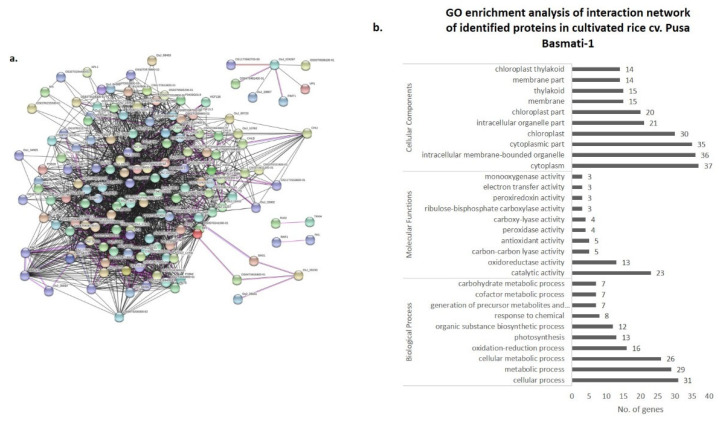
(**a**,**b**). Protein interaction network analysis of differentially expressed proteins (including primary interactors) in cultivated rice cv. Pusa Basmati-1 using STRING v.11.0. (**a**) Network was constructed at an interaction score cut off value of ≥0.700 with maximum number of primary interactors set to 500, including experimental evidence (pink lines), gene fusion (red lines), and co-expression (black lines). (**b**) Graph showing GO enrichment analysis of interaction network representing the no. of genes classified in respective GO categories (FDR corrected *p* value ≤ 0.01).

**Figure 7 bioengineering-09-00589-f007:**
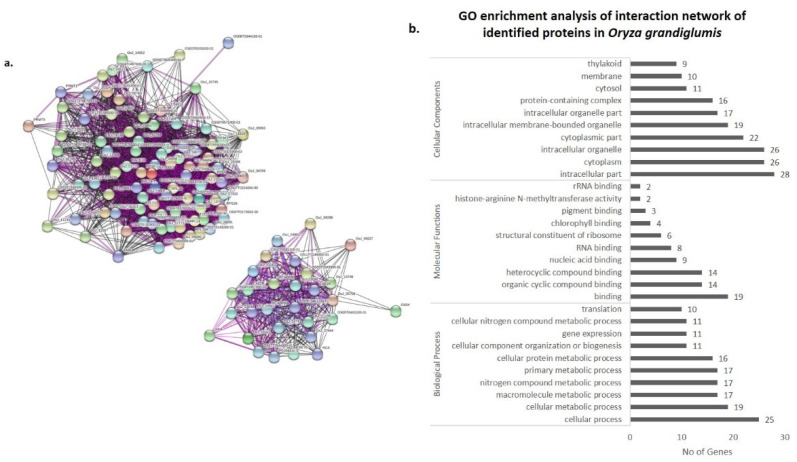
Protein interaction network analysis of differentially expressed proteins (including primary interactors) in wild rice accession *O.*
*grandiglumis* using STRING v.11.0. (**a**) Network was constructed at an interaction score cut off value of ≥0.700 with maximum number of primary interactors set to 500, including experimental evidence (pink lines), gene fusion (red lines), and co-expression (black lines). (**b**) Graph showing GO enrichment analysis of interaction network representing the no. of genes classified in respective GO categories (FDR corrected *p* value ≤ 0.01).

**Figure 8 bioengineering-09-00589-f008:**
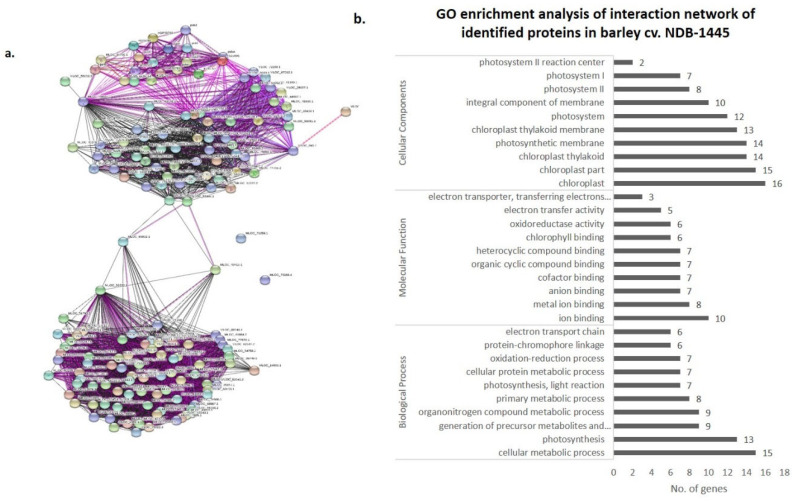
Protein interaction network analysis of differentially expressed proteins (including primary interactors) in barley cv. NDB-1445 using STRING v.11.0. (**a**). Network was constructed at an interaction score cut off value of ≥0.700 with maximum number of primary interactors set to 500, including experimental evidence (pink lines), gene fusion (red lines), and co-expression (black lines). (**b**) Graph showing GO enrichment analysis of interaction network representing the no. of genes classified in respective GO categories (FDR corrected *p* value ≤ 0.01).

**Figure 9 bioengineering-09-00589-f009:**
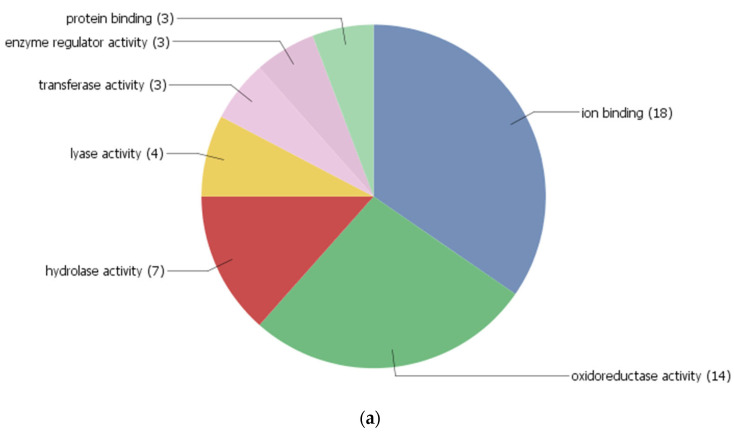
Classification of differentially expressed protein from cultivated rice cv. Pusa Basmati-1, wild rice accession *O. grandiglumis,* and barley cv. NDB-1445 proteins, based on molecular process and cellular component. (**a**) Pie chart showing functional categorization of identified proteins based on the molecular process. (**b**) Pie chart showing functional categorization of identified proteins based on the cellular component.

**Table 1 bioengineering-09-00589-t001:** List of proteins identified by the MALDI-TOF-MS/MS analysis of cultivated rice cv. Pusa Basmati-1 in response to *R. solani* inoculation.

Spot ID	Sequence ID	Gene Location	Protein Name	Theoretical M.Wt (KDa)/pI	SSR	Putative Molecular Function	Putative Biological Process	Predicted Sub Cellular Localization
URP1	LOC_Os10g21268.1	Chr10:10861158..10862712	Ribulos bisphosphate carboxylase large chain precursor	53.7/6.58	RM25277	Magnesium ion binding	Carbon fixation	Chloroplast
URP2	LOC_Os07g44430.1	Chr7:26548652..26549804	Thiredoxin peroxidase	24/5.97	RM22046	Peroxi redoxin activity	Oxidation–reduction process	Cytoplasm
URP3	LOC_Os11g10480.1	Chr11:5712641..5716288	Alcohol dehydogenase	40.98/6.20	RM26250	Zinc ion binding	Oxidation–reduction process	Cytoplasm
URP4	LOC_Os03g16050.1	Chr3:8841268..8843069	Fructose-1,6-bisphosphatase	43.6/5.0	RM14709	Phosphoric ester hydrolase activity	Carbohydrate metabolic process	Chloroplast
URP5	LOC_Os02g06300	Chr2:3143758..3152627	Translation factor	77.15/5.83	RM12492	GTPase activity	GTP binding	Chloroplast/Mitochondria
URP6	LOC_Os08g35310.1	Chr8:22277856..22279955	O-methyltransferase	38.46/5.64	RM23235	O-methyltransferase activity	Protein dimerization activity	Cytoplasm
URP7	LOC_Os06g39712.1	Chr6:23574363..23575246	OsClp10-Putative Clp protease homologue	24.74/4.64	RM20351	Serine-type endo peptidase activity	Proteolysis	Chloroplast
URP8	LOC_Os12g19381	Chr12:11262563..11278448	Ribulose bisphosphate carboxylase small chain	19.63/9.03	RM27952	Monooxygenase activity, copper ion binding	Photorespiration, photosynthesis	Chloroplast
URP9	LOC_Os07g08840	Chr7:4574245..4576234	Thioredoxin	13.15/5.17	RM21102	Protein disulfide oxido reductase activity	Cell redox homeostasis	Cytoplasm
URP10	LOC_Os03g53740	Chr3:30809029..30810138	Protein not annotated (Hypothetical protein OSJNBa0069E14)	18.551/5.0	RM15932	-	Systemic acquired resistance, salicylic acid mediated signaling pathway	Mitochondria
DRP1	LOC_Os03g36540.1	Chr3:20247674..20250146	Magnesium-chelatase subunit chlI	44.85/5.51	RM15319	Nucleoside-triphosphatase activity	Porphyrin-containing compound biosynthetic process	Chloroplast
DRP2	LOC_Os11g10480.1	Chr11:5712641..5716288	Alcohol dehydrogenase	40.984/6.20	RM26250	Zinc ion binding	Oxidation–reduction process	Cytoplasm
DRP3	LOC_Os03g03910.1	Chr3:1787648..1790957	Catalase domain containing protein	56.76/6.93	RM14344	Catalase activity, Heme binding	Oxidation–reduction process	Peroxisome
DRP4	LOC_Os10g21268.1	Chr10:10861158..10862712	Ribulose bisphosphate carboxylase large chain precursor	53.71/6.58	RM25277	Magnesium ion binding	Carbon fixation	Cytoplasm

**Table 2 bioengineering-09-00589-t002:** List of proteins identified by MALDI-TOF-MS/MS analysis of wild rice accession *O. grandiglumis* in response to *R. solani* inoculation.

Spot ID	Sequence ID	Gene Location	Protein Name	Theoretical M.Wt (KDa)/pI	SSR	Putative Molecular Function	Putative Biological Process	Predicted Sub Cellular Localization
URP1	LOC_Os03g11160.1	Chr3:5749954..5750298	Protein not annotated	12.39/5.04	RM14568	Cysteine-type endopeptidase inhibitor activity	defense response	Secreted
URP2	LOC_Os09g17740.1	Chr9:10845678..10847156	Chlorophyll A-B binding protein	28.01/5.14	RM24092	-	Photosynthesis	Chloroplast
URP3	LOC_Os12g39980.1	Chr12:24703185..24719302	Kinesin motor domain containing protein	316.6/5.0	RM28598	Microtubule motor activity	microtubule-based movement	Plasma Membrane
URP4	LOC_Os05g51830.1	Chr5:29753361..29756325	ZOS5-12-C2H2 zinc finger protein	32.54/4.53	RM19225	Metal ion binding	-	Nucleus
DRP1	LOC_Os04g31070.1	Chr4:18560177..18564417	Acyl carrier protein desaturase	43.34/6.53	RM16817	Acyl-[acyl-carrier-protein] desaturase activity	Oxidation–reduction process, Fatty acid metabolic process	Chloroplast
DRP2	LOC_Os10g28840.1	Chr10:15036094..15036951	Speckle-type POZ protein	30.60/5.57	RM25449	Protein binding	-	Chloroplast

**Table 3 bioengineering-09-00589-t003:** List of proteins identified by MALDI-TOF-MS/MS analysis in barley cv. NDB-1445 in response to *R. solani* inoculation.

Spot ID	Sequence ID	Gene Location	Theoretical M.Wt (KDa)/pI	Protein Name	Putative molecular function	Putative Biological Process	Predicted Sub Cellular Localization
URP1	MLOC_75166.4	7:256555607-256557498	41.8/5.4	Heat shock transcription factor	Sequence-specific DNA binding transcription factor activity	Responds to heat	Nucleus
URP2	MLOC_81876.1.1	2:600305903-600309626	53.1/4.8	Peptidase CIA, Pepain	Cysteine-type peptidase activity	Proteolysis	Vacuole
URP3	MLOC_78630.1	2:510753217-510754262	34.4/5.5	Photosystem II PsbO	Calcium ion binding	Photosynthesis	Chloroplast
URP4	MLOC_52332.1	7:136313164-136314932	35.5/4.4	Ribonucleotide reductase small subunit	Deoxy ribonucleoside diphosphate metabolic process	Oxidation–reduction process	Cytoplasm
DRP1	MLOC_945.1	5:524670853-524671657	28.8/5.6	Chlorophyll A-B binding protein	Chlorophyll binds, Metal ion binding	Photosynthesis, light harvesting	Chloroplast
DRP2	MLOC_71259.1	7:500270477-500271934	49/5.7	--	Catalytic activity	--	Cytoplasm

## Data Availability

The original data were available from the corresponding author upon an appropriate request.

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
