# Peer review of "Proteo-Molecular Investigation of Cultivated Rice, Wild Rice, and Barley Provides Clues of Defense Responses against Rhizoctonia solani Infection"

_bioengineering, 2022, doi:10.3390/bioengineering9100589_

Round 1
Reviewer 1 Report
Dear authors
Your article is very interesting but you should revise it before resubmit again to this Journal
extensive reading by English-native reader is necessary for a suitable publication in this international journal
it is better to added good recommendation in abstract
Also write your objective clear
how many replicate you used in your experiment
what the size of pots you used
it is not clear for me how you inoculated the plants
what the type if statistical you used
put the names of the author(s) of all taxa cited the first time they appear in the text
Add DOI for all your references.
The style and font of all references should be the same.
conclusion should rewrite again and focus on your best results
Author Response
Dear Sir,
Good day
All the comments and suggestions raised by the estimated reviewer are now incorporated in the manuscript.
Q 1 Extensive reading by English-native reader is necessary for a suitable publication in this international journal
Ans. The manuscript is now corrected as per the need of journal.
Q 2 It is better to added good recommendation in abstract
Ans.: Abstract is now corrected
Q.3. Also write your objective clear
Ans.: Now clear objective is mentioned in the manuscript
Q.4. how many replicate you used in your experiment
Ans.: Now three replication in the manuscript is mentioned in the appropriate places.
Q.5. what the size of pots you used
Ans.: Size of pots is now mentioned in the materials and methods section.
Q.6. it is not clear for me how you inoculated the plants
Ans.: 9 plants was used for the inoculation purpose and it is now mentioned in the section
Q.7. what the type if statistical you used
Ans. CRD and student t test was used and now mentioned in the manuscript.
Q 8. Put the names of the author(s) of all taxa cited the first time they appear in the text
Ans. Now mentioned
Q.9. Conclusion should rewrite again and focus on your best results
Ans. Now conclusion section is shortened with the best result

Reviewer 2 Report
The MS describes an important disease of rice and proteo-molecular investigation; however, it is filled with several technical and grammatical errors. My major concern is about data analysis. Results given should be supported with statistical value that is major drawback of the paper. Although, the bioinformatics part is ok with me, but I have never seen in the MS where authors mentioned the repetition of the experiments, number of samples taken for analysis, that is very important to remove biological errors. Although authors got culture from Plant pathology division, but what was the guarantee that it is pure culture, can author mention NCBI gene sequence number of the isolate that they used to support this statement. Further, after getting isolates why did they not conduct pathogenicity test to confirm its virulence. Further, susceptible and resistant check is also missing in experiment with disease severity. On what basis they collected sample for the enzymatic and proteomic analysis. These all are major issue that should be corrected before publication. I have mentioned several others comments that authors can find useful in revision.

Author Response
As per the suggestions made in the pdf copy of the manuscript, most of the corrections have been incorporated. Figure 1 is now replaced with new pics. Figure 6 and figure 7 are now deleted from the main manuscript and added in the supplementary figure section. Now corrections from the abstract to the conclusion are incorporated in the manuscript. References also decreased from 99 to 62.

Round 2
Reviewer 2 Report
Authors addressed by comments however, still some mistakes have to be corrected
Line 57 hexaconazole, propiconazole are systemic fungicides not non-systemic, correct
Line 108 was carried out
Line 139 except R. solani other are not italic
Line 708 references next
Still data analysis part is not clear to me as I mentioned in my previous comments
Author Response
- Now, systemic fungicides are replaced with non-systemic fungicides.
- Now line 108 is corrected.
- Now italic words are corrected.
- References section is corrected
- Now data analysis part is modified
